# Prader–Willi Syndrome: Possibilities of Weight Gain Prevention and Treatment

**DOI:** 10.3390/nu14091950

**Published:** 2022-05-06

**Authors:** Éva Erhardt, Dénes Molnár

**Affiliations:** 1Department of Paediatrics, Medical School, University of Pécs, H-7623 Pécs, Hungary; molnar.denes@pte.hu; 2National Laboratory for Human Reproduction, Szentágothai Research Centre, University of Pécs, H-7623 Pécs, Hungary

**Keywords:** Prader–Willi syndrome, nutritional phases, obesity, treatment, prevention

## Abstract

Prader–Willi syndrome (PWS) is a complex genetic disorder which involves the endocrine and neurologic systems, metabolism, and behavior. The aim of this paper is to summarize current knowledge on dietary management and treatment of PWS and, in particular, to prevent excessive weight gain. Growth hormone (GH) therapy is the recommended standard treatment for PWS children, because it improves body composition (by changing the proportion of body fat and lean body mass specifically by increasing muscle mass and energy expenditure), linear growth, and in infants, it promotes psychomotor and IQ development. In early childhood, the predominant symptom is hyperphagia which can lead to early onset, severe obesity with different obesity-related comorbidities. There are several studies on anti-obesity medications (metformin, topiramate, liraglutide, setmelanotide). However, these are still limited, and no widely accepted consensus guideline exists concerning these drugs in children with PWS. Until there is a specific treatment for hyperphagia and weight gain, weight must be controlled with the help of diet and exercise. Below the age of one year, children with PWS have no desire to eat and will often fail to thrive, despite adequate calories. After the age of two years, weight begins to increase without a change in calorie intake. Appetite increases later, gradually, and becomes insatiable. Managing the progression of different nutritional phases (0–4) is really important and can delay the early onset of severe obesity. Multidisciplinary approaches are crucial in the diagnosis and lifelong follow-up, which will determine the quality of life of these patients.

## 1. Introduction

Prader–Willi syndrome (PWS) is a complex genetic disorder that develops through different clinical phases and progresses by age. PWS affects all genders equally and presents in 1/10.000–1/30.000 individuals [1]. The majority of PWS cases are sporadic, and it was the first genetic syndrome attributed to genomic imprinting. The underlying genetic abnormality is caused by the loss of paternally expressed genes on the ‘critical region’ of chromosome 15q11–q13. There are three molecular mechanisms: paternal deletion of the 15q11–q13 (65–75%), maternal uniparental disomy 15 (20–30%), and imprinting defects (1–3%) [2]. The typical challenges of the syndrome are related to: (1) nutrition, from poor appetite to the development of hyperphagia. The former causes failure to thrive (FTT), and the latter leads to severe obesity; (2) persistent muscle hypotonia and reduced muscle mass; (3) behavioral aspects characterized by rigidity, obsessive-compulsive behavior, temper tantrums, and the need for predictability and a structured daily routine. To consistently encourage, implement and maintain a strict diet and daily physical activity while dealing with behavioral problems is undoubtedly challenging for children with PWS and their caregivers. Those afflicted with the disorder also have a significantly increased risk of developing obesity-related disorders and increased mortality, mental disorders, orthopedic problems, sleep problems, and various endocrinological complications [3].

Severe obesity is one of the most important symptoms of PWS, with a prevalence of around 40% in children and adolescents [4], and it is increasing in adulthood. Obesity occurs through different mechanisms, such as hypothalamic abnormalities of satiety, growth hormone deficiency, functional disorders of different hormones taking part in food intake regulation, and slowing down the movement of the gastrointestinal tract. Because of reduced muscle mass, the energy expenditure (EE) is decreased, although there is a normal relationship between fat-free mass and resting EE (REE). Sleep disorders with reduced REM and central as well as obstructive apnea are also risk factors for developing severe obesity. Patients with PWS have dysregulation at the level of the hypothalamus of the orexigenic neuropeptide Y (NPY), agouti-related protein (AgRP), and gamma-aminobutyric acid (GABA) neurons as a consequence of the deletion of SNORD116 in the PWS critical region [5,6]. Moreover, in patients with PWS, the circulating levels of acyl-ghrelin and leptin are increased, whereas peptide YY (PYY) levels are reduced. Insulin resistance is also typical. These hormonal changes result in the decreased production of hormones promoting satiety in the hypothalamus [7,8].

The aim of this paper is to summarize current knowledge on the dietary management and treatment of PWS, mainly to prevent excessive weight gain.

## 2. Methods

Published literature was retrieved through the electronic searches of Medline, PudMed, and Foundation of Prader–Willi syndrome databases in February 2022, using appropriate controlled keywords and vocabulary (e.g., Prader–Willi syndrome, obesity, pharmacological treatment of PWS, nutritional treatment of PWS, dietary intake in PWS, treatment/therapy in PWS, anti-obesity drugs or ‘drug name’). Results were screened for articles presenting data on current standard therapy in PWS.

## 3. Clinical Aspects of Prader–Willi Syndrome

The consensus for diagnostic criteria to support the diagnosis of PWS was developed by Holm, Cassidy, Butler et al. in 1993 [9]. Although these criteria help the diagnosis of PWS, the final, conclusive diagnosis requires molecular genetic testing [1].

In clinical practice, many features in patients with PWS can be subtle and non-specific; however, there are some primary characteristics that may be observed in most cases. Although certain features are characteristic of patients with PWS and remain unchanged throughout the disorder, there are some signs and symptoms that appear in varying degrees depending on the patient’s age. These periods are the following: prenatal and newborn, infancy, early and late childhood, adolescence, and adulthood [10].

The other possibility to classify the clinical characteristics of PWS is according to nutritional phases. Classically, two main nutritional development stages have been described: Stage 1, where the patients have feeding problems, muscle hypotonia, and FTT in infancy, followed by Stage 2 in childhood, in which the main characteristic is hyperphagia leading to obesity [11,12]. It is known that a more complex progression of the phases characterizes the syndrome.

In the study of Miller et al. [12], four main phases are identified and further divided into sub-phases. Obesity develops in the second year of life, after an initial phase of poor feeding. A gradual shift occurs over several nutritional phases while the syndrome develops progressively. Decreased fetal movements and growth restriction are typical during intrauterine life (phase 0). In early infancy (phase 1a), muscle hypotonia, poor feeding, and FTT leads to feeding problems and reduced appetite initially (0–9 months of age); from 9 to around 24 months of life, both food intake, appetite, and weight gain return to normal (phase 1b). Children begin to gain weight from the age of 2 years (phase 2a) without a change in appetite, whereas in Phase 2b, an increased interest in food is observed, and patients experience excessive weight gain due to a marked increment in appetite (4.5–8 years). Hyperphagia, insatiable appetite, and obesity are typical characteristics by school age (median age of onset: 8 years) and increase during childhood (phase 3) through to adulthood. However, the majority of adult patients develop satiety, and food-seeking behavior may improve (phase 4).

## 4. Nutritional Aspects in Weight Gain Prevention

Nutritional management by nutritionists and gastroenterologists plays a major role in weight control in children with PWS, and specifically in terms of dietary intake. Appetite behavior in these children is a life-long challenge and can be life-threatening. The appetite of PWS patients changes [12,13] during the different phases of nutrition. In Phase 1a, children with PWS have no desire to eat and often will have FTT in infancy despite adequate calories (0–9 months). The nutritional goal in this period is to promote optimal growth without developing obesity. Decreased muscle mass and increased fat mass are present from infancy. Infants with PWS require less energy intake compared to healthy children. The physiologic background of this phenomenon is the reduced REE and reduced energy spent on physical activity than in healthy infants [12,14]. In the Phase 1b period, the appetite and weight return to normal, and generally continues up until the age of 36 months. Phase 2a begins at age 18–36 months, where weight increases without any change in appetite, and lasts until approximately age 4.5. During this period, appetite is appropriate for age. Gastric motility appears to slow down, and metabolism begins to change. Children with PWS will become obese if the recommended daily allowance (RDA) of calories is given at this age. The REE is decreased in these children and approximately 50–60% of the RDA is sufficient in this phase. An increasing interest in food begins between 4.5–8 years of age (in phase 2b). Appetite increases gradually and becomes insatiable [15,16]. The average onset of Phase 3 is around 8 years of age when an insatiable appetite develops. The weight gain can be rapid and life-threatening and may occur even on a low-calorie diet. The guidelines recommend 10–12 kcal/cm of height for weight maintenance and 6–8 kcal/cm for weight loss in pre-school- and school-aged children [17,18,19,20]. Children with PWS must be supervised where food is accessible at all times. In adulthood (Phase 4), the appetite is satiable and can decrease in rare cases.

Irizarry et al. investigated [21] eight children who were randomized to consume either low-carbohydrate, high-fat (LC, 15% carb; 65% fat; 20% protein) or low-fat, high-carbohydrate (LF, 65% carb, 15% fat, 20% protein) diets. They examined the effect of these diets on hormonal and metabolic changes of these children with PWS. Those subjects who consumed the LC diet had lower postprandial insulin, higher fasting glucagon-like peptide 1 (GLP-1) concentrations, increased postprandial GLP-1 and a reduced fasting ghrelin to GLP-1 ratio, compared to those on a LF diet. No significant weight changes occurred during the study.

Miller et al. [22] examined the effect of an energy-restricted diet with modified macronutrient distribution on body weight in children with PWS aged 2–10 years. 33 children of 63 participants consumed the recommended diet, which was composed of 30% fat, 45% carbohydrates, and 25% protein, with at least 20 g of fiber per day. They found that children on the calculated diet displayed significant improvements in body fat (*p* < 0.0001) and weight (*p* < 0.001) with a lower respiratory quotient (RQ) (*p* = 0.002) than those on a diet with a reduced energy intake.

Until there are specific treatments for hyperphagia/metabolism, there is a need to control weight with the help of traditional tools, diet and exercise. Recently, the ’best’ diet has been reported [22] with a composition of 45% or less complex carbohydrates (up to 20 g fiber/day) 30% fat, and 25% protein, which works best for weight control and improvement in body composition.

Many studies have been published [17] on diet restriction and physical activities (PA) in children with PWS, sometimes with contradictory results. PA decreases in children with PWS because of obesity, somnolence, and persistently decreased muscle tone. The altered body composition of these patients also contributes to a reduction in EE. Increased PA and exercise programs are beneficial in improving body composition. Supervised PA programs are important in children with PWS, and these programs should be planned individually.

## 5. Pharmacologic Treatment Options

### Growth Hormone Therapy

Patients with PWS have hypothalamic dysfunction, which may lead to several endocrinopathies such as hypothyroidism, hypogonadism, central adrenal insufficiency, and growth hormone deficiency (GHD). All individuals with PWS should be considered growth hormone (GH) deficient. The prevalence of GHD varies 40–100% based on different studies [23]. There was a 74% prevalence of GHD in the KIGS (Kabi International Growth Study) database [24]. This high discrepancy in prevalence is likely to be due to the variation of the testing methods, differences in GH assays, and the patients’ age at the time of testing. The exact pathogenesis of GHD is not clear. The aim of GH therapy is to improve body composition (decrease fat mass, increase muscle mass) and promote growth. However, IQ also improves with GH therapy, especially in children with severe cognitive delays. Carrel et al. [25] demonstrated that GH treatment in infants improves IQ and psychomotor development, so it is recommended to start GH therapy as soon as possible (often around 3–6 months of age). Miller et al. revealed [22] that children treated with GH before the first year of life had decreased fat mass and higher REE compared to those treated after one year of age, which may indicate that early treatment with GH may help ameliorate obesity in PWS. GH dosing needs to be individualized based on IGF-1, growth velocity, and body composition/height.

Short-term treatment with growth hormones for 1–2 years may not normalize body weight but significantly increases lean body mass and decreases fat mass. In addition, height in relation to age may increase and psychomotor development and behavior may improve in young children [26,27,28].

A 4-year-long, continuous GH treatment [29] in prepubertal children with PWS led to a pronounced improvement in body composition, but complete normalization could not be achieved. Why normalization could not be achieved may be due to the fact that the typical body composition in PWS is not only due to growth hormone deficiency but also a genetic defect affecting other pathways.

In another study, patients began GH therapy at the age of 3.7 years and continued to receive it for 8 years [8]. Body composition significantly changed in the first year of therapy, whereas this tendency slowed down in subsequent years and by the end of the 8-year treatment period, the body mass index (BMI) values of subjects with PWS remained higher than in normal children [30].

Contraindications of GH treatment in patients with PWS are severe obesity and severe obstructive sleep apnea [23,31].

PWS cannot be prevented, and there is no remedy for it. GH replacement treatment is the only Federal Drug Agency (FDA)-approved treatment for PWS [5]. Weight control is one of the most important goals in PWS therapy. However, it is a challenging task due to behavioral problems related to appetite control, decreased lean body mass, and reduced REE.

There are no widely accepted guidelines on the acceptability, safety, or efficacy for anti-obesity medication (AOM) in children with PWS. Table 1 shows medications that may be used for reducing weight and hyperphagia in patients with PWS.

Goldman et al. published a literature review [6] and case series on drugs used in children and adolescents with PWS. 14 articles (3 topiramate, 1 metformin, 4 liraglutide, 5 oxytocin, 1 naltrexone–bupropion) were yielded from their literature search. All studies reported enhanced hyperphagia with variable BMI effects. No significant side effects were found.

***Topiramate*** is an antiepileptic drug with off-label use for weight reduction. Its mechanism is not totally understood. Topiramate appears to block sodium and calcium channels, increasing the suppression effects of GABA. A double-blind, randomized placebo-controlled eight-week study was published [37] with a trend in reducing BMI without statistical significance, and a dose-dependent raise in hyperphagia among 62 patients (ages 12–45 years). An open-label study in seven patients with PWS demonstrated weight reduction or declined weight gain and improved mood [38]. An 11-year-old male with PWS was described whose aggression and ’demand for food’ decreased following treatment with topiramate [39].

***Metformin*** is an oral drug classically used in type 2 diabetes to improve hyperglycemia and is also used off-label to treat obesity and prediabetes [40]. Effects of metformin are multi-factorial, including reduced glucose production in the liver and glucose absorption in the gastrointestinal tract; thus, insulin sensitivity increases. [34].

In a pilot study metformin treatment was started in 21 children and adolescents with PWS who had insulin resistance and glucose intolerance. Metformin treatment improved food-related distress and anxiety, evaluated by Hyperphagia Questionnaire; however, weight reduction did not occur. [34]. A total of 7 out of 10 males had to stop metformin due to a worsening behavioral condition.

***Naltrexone-bupropion*** is a combined drug used to treat obesity and impulsive behavior. Naltrexone is an opioid receptor antagonist, and bupropion is a dopamine/norepinephrine reuptake inhibitor [6]. It has a role in the hypothalamus through the anorexigenic effect of the α-melanocyte-stimulating hormone from pro-opiomelanocortin neurons. [32,41]. The FDA has approved this drug for the treatment of obesity [6,40]. From the literature search, one study investigated the effect of this therapy in 13-year-old children with PWS. A decreased BMI with improved aggression was reported [32].

***Glucagon-like peptide-1 (GLP-1) receptor agonist*** stimulates insulin release, inhibits glucagon secretion, and decreases plasma ghrelin levels. According to a systematic review on PWS patients, gastrointestinal tract motility decreased, and satiety increased after eating [35]. Liraglutide is the short-acting, daily preparation approved by the FDA and EMA for the treatment of obesity in patients older than 12 years [6,40].

In a review [33], PWS patients who also had type 2 diabetes were treated with 1.2 to 1.8 mg/day of liraglutide (4 patients) and 20 mg/day of exenatide (2 patients). After 24 months of treatment, a tendency of reduction in BMI, HbA1c, waist circumference and improved glycaemia was seen [20,33]. A longitudinal study evaluated the efficacy of exenatide therapy (6 months) in overweight or obese young adults with PWS. Reduced appetite and an improvement in HbA1c were reported, whereas weight or BMI did not change. [36].

***Oxytocin*** is a neuropeptide that has a wide range of indications. It induces weight reduction by increasing EE and lipolysis, as well as diminishing appetite, which can lead to decreased food intake [6]. Five studies investigated the effect of the intranasal administration of an oxytocin analogue (carbetocin) in children with PWS. Three studies showed improvement in social and food-related behaviors [42,43,44], whereas two studies could demonstrate only limited positive effects on behavior. [45,46].

Other drugs targeting weight control are under investigation as potential treatments for PWS. Tesofensine is a triple monoamine reuptake inhibitor of the neurotransmitter’s dopamine, norepinephrine, and serotonin that acts as an appetite reducer and has been connected with weight loss in a placebo-controlled trial of patients with obesity in Denmark [8,47].

Another drug is diazoxide, an ATP-sensitive K+ channel agonist that inhibits insulin secretion from the pancreas and modulates insulin-sensitive enzymes, leading to suppressed lipogenesis and increased lipolysis. It has been shown that diazoxide decreases fat mass, weight, and blood glucose levels in a mouse model with PWS [48]. A pilot study assessing the efficacy and safety of diazoxide choline found a significant decrease in fat mass and a significant reduction in hyperphagia among nine patients with PWS [49]. More extensive studies are in progress to substantiate the use of diazoxide choline in adolescents and young adults with PWS.

Setmelanotide (or RM-493) is an agonist of the appetite-regulating melanocortin-4 receptor, which is being examined for different forms of genetic obesity [17].

There is an unacylated ghrelin analog that has been shown to inhibit the orexigenic effect of unacylated ghrelin in animals, and it is assumed to have favorable metabolic effects in humans [50].

## 6. Prevention of Obesity in Prader–Willi Syndrome

In general, the prevention of obesity in childhood is a major task for pediatricians. In children with PWS, it is important to introduce preventive programs as early as possible to control caloric intake, which is very difficult due to the low compliance of these children.

Preventive methods for weight management in PWS must be performed simultaneously by using dietary, physical activity and behavioral interventions. The role of a multidisciplinary team is mandatory to treat these children. During the first months, the main task of nutritionists is to ensure the appropriate caloric intake of children with PWS [17,51]. After the first year of life, a low-calorie but well-balanced diet is required, and close supervision is needed to minimize food-stealing [20]. Physical activity and the encouragement of muscle mass training has to be a daily routine in subjects with PWS at all ages. [52]. PA level is decreased in PWS patients due to low (skeletal) muscle mass and muscle hypotonia compared to healthy children [52]. Complex and early interventions are recommended to promote skill acquisition and improve motor development [20].

Bellicha et al. published [53] a systematic review on habitual PA and sedentary behavior and the effects of PA in children with PWS. The effect of eight different interventions were explored. The summarized benefit of PA interventions was due to the improvement of physical fitness, such as improved walking capacity, muscle strength, and gait parameters. In contrast, only one study showed a significant weight reduction after PA intervention, whereas the others did not find significant weight and/or fat loss. In subjects with PWS, PA interventions seem to have a beneficial effect on lean body mass [54] and bone mineral density [55]. Observational studies have also reported a positive relation between habitual PA and lean body mass [53,54] or bone parameters [56].

GH treatment could also have an important role in the prevention of obesity, when initiated during infancy. Patients with PWS greatly require behavioral programs in which, besides medical personnel, teachers, friends, parents, and other family members are also to be included. Supervised PA programs are feasible in children with PWS and these programs should be arranged individually and should offer a variety of enjoyable activities [20,53].

## 7. Discussion

PWS is an exceptionally complex disorder, both genetically with different molecular mechanisms that produce several different phenotypes and with a multifaceted clinical development through several phases.

The different genetic mechanisms in PWS that lead to different phenotypes can give different variations in the challenges of the syndrome. For example, children with PWS, caused by uniparental disomy will often have a lower incidence of behavioral problems, but a significantly increased risk of autism spectrum disorders and affective disorders, including psychoses. On the other hand, in the case of a chromosomal deletion, especially type I deletion, severe behavioral problems are more frequent [57].

Lifelong diet restriction is an important aspect of PWS treatment [17]. In order to maintain the diet, many factors are to be under control, such as strict habits, limitation of leniency for children to make their own decisions regarding food, by following a fixed nutritional program instead. The main mechanisms and cause of hyperphagia in PWS is still uncertain. It will be important to investigate this more closely in the future, in order to further develop pharmaceutical agents that can help reduce the extent of the hyperphagia. Perhaps such drugs could also contribute to a reduction in unwanted ill attitudes and temper tantrums, which are often associated with situations involving food in children. By reducing hyperphagia, one may be able to prevent the development of obesity, one of the main factors leading to the high mortality rate of individuals with PWS.

Early diagnosis, a strict diet regimen, physical activity, follow-up by multidisciplinary teams and growth hormone treatment have improved our understanding of the cognitive and physical issues in recent years, and helped improve the lives of many with PWS. Unanswered questions still exist related to the precise understanding of the relationship between genetic defects and the various phenotypes. Perhaps these unknown answers can contribute to further improvement in the care and treatment of children with PWS in the near future.

## Figures and Tables

**Table 1 nutrients-14-01950-t001:** Medication for the management of hyperphagia and obesity in PWS.

Drug	Indication/Approved Age	Mechanism of Action	Side Effect	Reference for PWS
**Sympathicomimetic**				
Phentermine	>16 years for management of obesity	amphetamine analog, increasing catecholamines and serotonin activity in CNS resulting appetite suppression	increased blood pressure, tachycardia	0 [6]
**Antiepileptic**				
Topiramate	>2 years for epilepsy >12 years for migraine off label for management of obesity	modulates Na+ channels, GABA agonist improves food-seeking behavior	fatigue, dizziness, mood changes, ataxia nephrolithiasis	3 studies [32,33,34]
**Opioid receptor antagonist/dopamine and** **noradrenaline reuptake** **inhibitor**				
Naloxone-Bupropion	for management of obesity	opioid receptor agonist + increase the POMC activity in the melanocortin system of hypothalamus, so decreases hunger and increases EE	high blood pressure, headache, insomnia, dry mouth, diarrhea, vomiting	1 case report [32]
**Pancreatic lipase inhibitor**				
Orlistat	>12 years for management of obesity	limiting fat absorption of up to 30% of ingested fats	gastrointestinal symptoms, liver injury	0 [6]
**GLP-1-R agonists**			gastrointestinal symptoms	
Liraglutide	>12 years for management of obesity	increases insulin secretion	nausea, delayed gastric emptying. tachycardia	4 case reports, 1 study [35,36]
Exenatide		increases insulin secretion	nausea, delayed gastric emptying. tachycardia	1 study [33]
Semaglutide (depot version)		increases insulin secretion	nausea/vomiting, diarrhea, constipation	0
**Insulin sensitizer**				
Metformin	>10 years for treating T2DM off label for management of obesity	improves insulin sensitivity in liver and muscle; anorectic effect; to increase GLP-1 from intestine	abdominal discomfort with diarrhea, nausea, vomiting lactic acidosis	1 pilot study [34] Case reports [6]
**Appetite/satiety modulator**				
Oxytocin	>12 years for management of obesity	modified of G-protein-coupled receptors, changing the production of PG, increasing EE and lipolysis, reducing appetite	tachycardia, gastrointestinal symptoms	5 intranasal studies [6]

Abbreviations: PWS: Prader–Willi syndrome; CNS: Central Nervous System, GABA: Gamma-aminobutyric acid; POMC: Pro-opiomelanocortin; EE: Energy Expenditure; T2DM: Type 2 Diabetes Mellitus; GLP-1: Glucagon-like peptide 1; PG: prostaglandin.

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
