# Peer review of "Prader–Willi Syndrome: Possibilities of Weight Gain Prevention and Treatment"

_nutrients, 2022, doi:10.3390/nu14091950_

Round 1

Reviewer 1 Report

Thank you for the opportunity to review this manuscript. This is an interesting topic that can be considered by readers. The manuscript is generally well prepared and covers a very important topic. However, after reviewing it, there is some point I think the authors should consider in order to improve this work:

  1. The authors describe various previous studies, outcomes and therapy guidelines. However, We do not know anything about the methodology of literature review (including how thoroughly the existing literature was searched - i.e. what keywords and what databases were searched?, what inclusion/exclusion criteria were used? etc.; what is the type of review article? etc.).

Author Response

Thank you very much for your work and time to asses my paper!

Please see the attachment below. 

Reviewer 2 Report

This is a concise review on the topic of weight manage ent in Prader-Willi Syndrome.

I have the following suggestions:

Line 51. Pluralize “mechanism” in “mechanisms”.

Line 53-55. This part of the long sentence is not so clear. Please, clarify it.

Lines 63 and 68. Please, put a “. “ at the end of the sentence.

Line 79. Please, write “the patients have” or “the patient has”, instead of “the patient have”.

Line 86. “progresses” instead of “progress”.

Line 88. After having acronymised previously “fallire to thrive” in “FTT”,  continue with the acronym a d do not go Back to the whole feature.

Line 116. “develops” instead of “developes”.

Line 117. Please, amend the punctuation in this sentence. “life-threatening and may occur even on low.calorie diet,” should be amended in “life-threatening, and may occur even on low calorie diet.”

Line e169. Delete the first “,”.

Line 173. Delete the space before the “,”.

LLine175. Delete the last “,”.

LLine 182. Delete the 2nd “subjects”.

Lines 192 and 266. Once more, after acronymizing a concept, the authors should continue with the acronym, not going back to the whole item. Then, here and afterwards, they should amend Prader-Willi Syndrome to PWS.

Line 244. Delete the space before the “,”.

Line 277. Delete the “,” after “Complex”.

Line 300. Put a “,” after “PWS”.

Author Response

Thank you very much for your precise and valuable feedback! 

All the mentioned mistakes, typing errors and spelling had been corrected. The new version of my manuscript is uploaded. 

Round 2

Reviewer 1 Report

Thank you.